# Towards a decision support tool for intensive care discharge: machine learning algorithm development using electronic healthcare data from MIMIC-III and Bristol, UK

Christopher J McWilliams,[1] Daniel J Lawson,[2] Raul Santos-Rodriguez,[1] Iain D Gilchrist,[3] Alan Champneys,[1] Timothy H Gould,[4] Mathew JC Thomas,[4] Christopher P Bourdeaux[4]

[1] Engineering Mathematics, University of Bristol, Bristol, UK
[2] Integrative Epidemiology Unit, Population Health Sciences, University of Bristol, Bristol, UK
[3] Department of Experimental Psychology, University of Bristol, Bristol, UK
[4] Intensive Care Unit, University Hospitals Bristol NHS Foundation Trust, Bristol, UK

**Correspondence to**
Dr Christopher J McWilliams;
chris.mcwilliams@bristol.ac.uk

## ABSTRACT

**Objective** The primary objective is to develop an automated method for detecting patients that are ready for discharge from intensive care.

**Design** We used two datasets of routinely collected patient data to test and improve on a set of previously proposed discharge criteria.

**Setting** Bristol Royal Infirmary general intensive care unit (GICU).

**Patients** Two cohorts derived from historical datasets: 1870 intensive care patients from GICU in Bristol, and 7592 from Medical Information Mart for Intensive Care (MIMIC)-III.

**Results** In both cohorts few successfully discharged patients met all of the discharge criteria. Both a random forest and a logistic classifier, trained using multiple-source cross-validation, demonstrated improved performance over the original criteria and generalised well between the cohorts. The classifiers showed good agreement on which features were most predictive of readiness-for-discharge, and these were generally consistent with clinical experience. By weighting the discharge criteria according to feature importance from the logistic model we showed improved performance over the original criteria, while retaining good interpretability.

**Conclusions** Our findings indicate the feasibility of the proposed approach to ready-for-discharge classification, which could complement other risk models of specific adverse outcomes in a future decision support system. Avenues for improvement to produce a clinically useful tool are identified.

## INTRODUCTION

Demand for intensive care unit (ICU) beds is rising at a time when the resource is constrained.[1] In order to optimise the allocation of this resource, patients should be discharged from the ICU as soon as they no longer require the specialist input provided there. The reduced ICU capacity caused by

### Strengths and limitations of this study

► Training data from multiple source domains is leveraged to produce general classifiers.
► The restrictive feature representation tested could be expanded to better exploit the richness of available data and boost performance.
► Our approach has the potential to streamline the discharge process in cases where patient physiology makes them clear candidates for a de-escalation of care.
► High-risk patients would require additional levels of decision support to facilitate complex discharge planning.

discharge delay can result in the delayed admission of patients requiring critical care.[2 3] Furthermore, patients remaining in the ICU after they are medically fit to leave are at risk of iatrogenic harm and may experience detrimental effects on physical rehabilitation and psychosocial well-being.[4]

The identification of individuals that are ready to leave ICU is a key component of patient flow through the hospital. At present this identification is a manual process, relying on physicians reviewing patients on a ward round at a standard point in time. There is a lack of formal guidance to inform discharge readiness and as such the process is sensitive to both the decision-making heuristics of individual clinicians and structural factors within the hospital.[5] A number of studies have looked to address this problem by attempting to standardise the discharge process.

In a scoping review of these studies Stelfox et al[6] noted that, while a range of tools have been developed to characterise discharge

readiness, most studies have been single-centre and have not conducted comparative evaluations of different tools.

Increasingly ICUs are using clinical information systems (CISs) to collect, store and display physiological data. The availability of such routinely collected patient data presents the opportunity to apply methods from data science, with the potential to transform healthcare in a number of ways.[7 8] Two particular avenues for development are the automation of simple tasks[9] and the implementation of decision support systems,[10] both of which could reduce the cognitive load of clinicians and free up scarce resource for tasks that require human expertise. This work considers the ICU discharge process, which has accessible data from routine collection and requires a simple but important binary decision that could benefit from an evidence-based approach. Indeed, several statistical models have recently been developed to predict the risk of adverse events following intensive care discharge.[11–15] Such risk models are invaluable tools for clinical decision making and, in the context of ICU discharge, can provide information with which to plan complex de-escalations of care. For example, patients deemed to be at high-risk of readmission may benefit from continued close monitoring,[16] since early detection of deterioration is a strong predictor of outcome.[17 18]

In our previous work on the psychology of clinical decision-making we have demonstrated the effectiveness of simple 'nudge' based interventions in changing clinical practice.[19–21] Building on this foundation we were motivated to develop a classifier to automatically flag patients that appear physiologically fit for discharge. The intention is that such a screening tool could streamline morning ward rounds by allowing staff to focus their attention on the most likely-dischargeable patients. The tool could also prompt clinicians to consider discharge decisions at other times of day, outside of normal rounds. In 2003 Knight proposed a set of nurse-led discharge (NLD) criteria[22] with a similar aim—to expedite discharge from a high-dependency unit by allowing nurses to discharge patients who were clearly well enough to leave. These criteria represent a general and highly conservative set of constraints on physiology that characterise a patient as suitable for care on an acute ward (level 1 care). High-risk patients are unlikely to meet these criteria, but may still be dischargeable by a consultant. In this study we use routinely collected patient data to retrospectively evaluate Knight's criteria, and then improve on their performance using machine learning methods. To this end we study two historical cohorts. One cohort consists of patients treated on the general intensive care unit (GICU) at the Bristol Royal Infirmary between January 2015 and April 2017, while the second consists of patients selected from the Medical Information Mart for Intensive Care (MIMIC)-III database[23] (see Methods section for details).

## METHODS

### Discharge criteria

The NLD criteria proposed by Knight[22] consist of a set of constraints on various routinely collected vital signs and laboratory results. If a patient meets all the criteria for a period of at least 4 hours, Knight states that they may be safely discharged by a nurse. The motivation behind developing these criteria was to facilitate discharge by nurses in cases where the decision was clear, and there is some evidence of improved bed allocation when using such a nurse-led system.[22 24 25] In order to test the NLD criteria on historical patient data we codified the constraints (see online supplementary file section A) into 15 binary tests, which are defined in table 1. For criteria that were not assigned numeric values in the original publication (B1-4, central nervous system) we used the 'normal' bounds as defined in our CIS.

### Cohort selection

Subjects for this study were selected from two distinct historical data sources to form two patient cohorts. The inclusion criteria are detailed in online supplementary file section B. The first data source consists of the routinely collected data from 1870 patients treated on the GICU at the Bristol Royal Infirmary. We refer to the cohort selected from this dataset as *GICU*. The second data source was derived from the MIMIC-III database,[23] from which we selected patients who were admitted to medical or surgical intensive care since this approximates the patient type in GICU. We restricted our analysis to the 'Metavision' subset of MIMIC-III, since the labelling of the variables required to evaluate the NLD criteria was found to be more consistent in this portion of the database. Furthermore, we selected only the first intensive care stay of any given hospital admission, and only those stays for which there was a recorded *callout* (ready-for-discharge [RFD]) time. Following these criteria, we arrived at a subset of 7592 patients from MIMIC-III, forming the cohort we refer to hereafter as *MIMIC*.

The use of two cohorts was motivated by two concerns. First, by including the MIMIC cohort, we significantly increased the volume of data available for training classifier algorithms. Second, the use of two cohorts allowed us to study the generalisation of our results between different patient populations under different healthcare systems.

### Readiness-for-discharge

The key to testing and improving on the discharge criteria was to be able to identify, from the historical data, patients that were RFD and not-ready-for-discharge (NRFD). Whereas previous models have looked to predict the occurrence of adverse events following ICU discharge[12 15] we wanted to learn to classify those patients that appear physiologically fit to leave the unit. These are subtly different tasks. The former requires the identification of patients at risk of negative outcomes from those who have already been declared fit for discharge, while

**Table 1** Codified version of the discharge criteria for application to electronic health record data. Here the 15 criteria have been grouped into intuitive subsets and each assigned a test ID ('R0' to 'B4'). According to the original specification, if all 15 criteria are met for a period of at least 4 hours the patient can be safely discharged

| Test ID | Test name | Variable | Test condition |
|---|---|---|---|
| R0 | Respiratory: airway | airway | airway patent |
| R1 | Respiratory: Fio$_2$ | fio2 | fio2≤0.6 |
| R2 | Respiratory: blood oxygen | spo2 | spo2≥95 (%) |
| R3 | Respiratory: bicarbonate | hco3 | hco3≥19 (mmol/L) |
| R4 | Respiratory: rate | resp (rate) | 10≤resp≤30 (bpm) |
| C0 | Cardiovascular: blood pressure | bp (systolic) | bp≥100 (mm Hg) |
| C1 | Cardiovascular: heart rate | hr | 60≤hour≤100 (bpm) |
| P | Pain | pain | 0≤pain≤1 |
| CNS | Central nervous system | gcs | gcs≥14 |
| T | Temperature | temp | 36≤temp≤37.5 (°C) |
| B0 | Bloods: haemoglobin | haemoglobin | haemoglobin≥90 (g/L) |
| B1 | Bloods: potassium | k | 3.5≤k≤6.0 (mmol/L) |
| B2 | Bloods: sodium | na | 130≤na≤150 (mmol/L) |
| B3 | Bloods: creatinine | creatinine | 59≤creatinine≤104 (umol/L) |
| B4 | Bloods: urea | bun | 2.5≤bun≤7.8 (mmol/L) |

CNS, central nervous system.

the later looks to identify, from a sample of ICU patients, those who are no longer in need of critical care. Clearly the latter is an easier task. In order to train a classifier for this task it was necessary to define instances of the positive (RFD) and negative classes (NRFD). Both datasets (GICU and MIMIC) contain a callout for each patient, which marks the time at which a patient was declared clinically ready to leave the ICU. A patient was defined as RFD at their time of callout, provided they had a positive outcome after leaving ICU. Conversely, patients with a negative outcome were defined as NRFD at their time of callout. A positive outcome was defined as the patient leaving hospital alive without readmission to ICU. A negative outcome was defined as either readmission to

ICU during the same hospital admission, or in-hospital mortality after discharge from ICU. We note that it is more conventional to use readmission (or mortality) within 48 hours to define a negative outcome related to ICU care.[12 26] However, this practice is not universal[27] and in our case it was not possible because of limitations in the data available locally.

Given the low rates of negative outcome following callout in both MIMIC and GICU (see table 2), we generated further instances of the negative class, in order to balance the class sizes. Conceptually this is equivalent to providing more instances for the classifier to learn the physiological characteristics of patients requiring ongoing critical care. To do this we sampled patients

**Table 2** Patient characteristics for the two cohorts. Discharge delay defined as length of time between callout and discharge from intensive care unit (ICU). Readmission to ICU defined as readmission during same hospital stay. Negative outcome is in-hospital mortality and/or readmission

| | MIMIC | GICU |
|---|---|---|
| Total patients | 7592 | 1870 |
| Gender, % female | 47.6 | 40.5 |
| Age, median years (IQR)) | 64.0 (50.9–77.0) | 63.0 (49.0–72.8) |
| BMI, median (IQR) | 28.1 (24.9–31.6) | 26.5 (22.8–30.6) |
| Length of stay, median days (IQR) | 1.93 (1.11–3.34) | 2.96 (1.69–5.14) |
| Discharge delay, median days (IQR) | 0.27 (0.18–0.39) | 0.34 (0.20–1.04) |
| In-hospital mortality, # (%) | 466 (6.14) | 67 (3.58) |
| Readmission to ICU, # (%) | 589 (7.76) | 52 (2.78) |
| Negative outcomes, # (%) | 954 (12.57) | 109 (5.83) |

BMI, body mass index; GICU, general intensive care unit.

at between 3 days and 8 days prior to their callout (see online supplementary file section B: figures 1–3), under the assumption that patients were NRFD at this point in time, regardless of their eventual outcome state (positive or negative). Patients within the first 24 hours of their ICU stay were omitted from this sample. Full details of the sampling procedure are given in online supplementary file section B.

### Feature extraction

We used the same feature set to evaluate the NLD criteria and to train machine learning classifiers. We constructed either one or two features corresponding to each of the NLD criteria, depending on the criteria in question and on data availability. For example, the features 'resp min' and 'resp max' were used to test the criterion R4, whereas the single feature 'bun' was used to test B4. Where possible the feature values were calculated from a 4 hours sample window, as specified by the original NLD criteria. In the cases where no data was available during the 4 hours window, an extended 36 hours window was used. This extended window was mainly relevant for infrequently measured laboratory test results (see online supplementary file section C: table 1). Full details and justification of the feature extraction procedure are provided in online supplementary file section C. Since this feature set is somewhat restrictive, consisting of 18 physiological features, we also defined an *extended* feature set that included the following extra features: *age, sex, body mass index (BMI)* and *hours since admission.*

To produce the results presented in the main text, missing feature values were imputed using k-nearest neighbour imputation.[28] Full details of the imputation procedure are given in online supplementary file section D, along with a complete case analysis that addresses the sensitivity of our results to this imputation strategy. When training and testing the machine learning classifiers, features were standardised by subtracting the mean and dividing by the SD. The feature matrices for the imputed and complete case data sets are visualised using the t-Distributed Stochastic Neighbour Embedding (t-SNE) algorithm[29] in online supplementary file section D: figures 4 and 5.

### Analysis of NLD criteria

Knight originally specified that all 15 criteria must be met in order to allow safe discharge by a nurse.[22] Following this specification, we evaluated the criteria for both MIMIC and GICU, determining which instances were classified as RFD and NRFD, and comparing these results to ground-truth. We then further investigated the performance of the NLD criteria as a classification system, by relaxing the constraint that all 15 tests must be passed in order to make an RFD classification. Instead we used the NLD criteria to produce probability estimates of being RFD, by summing the number of tests passed and dividing by 15 to produce a normalised output between 0 and 1. In this formulation each of the 15 criteria contribute equally to

the RFD probability. Subsequently we weighted each of the criteria according to a measure of feature importance (see below) in order to improve their predictive performance. Using the probability outputs, it was possible to evaluate the performance of the NLD criteria in the same way as the machine learning classifiers described below.

### Machine learning classifiers

To improve on the performance of the NLD criteria, we trained and tested two machine learning classifiers: a random forest (RF)[30] and a logistic classifier (LC).[31] These two algorithms were chosen for their simplicity in implementation and ease of interpretation in their predictive output. The training methodology we used was intended to produce classifiers that made good use of the training data that comes from multiple source domains, while generalising well to new patient populations. As such we employed multiple-source cross-validation.[32] A single iteration of this procedure is as follows. Each source dataset is split into train and test data. For GICU 30% of the data is held out for testing. For MIMIC an equal sized test set is held out (~10%). Multiple-source cross-validation is then used to optimise the hyper-parameters on the training data (see online supplementary file section E) with two folds, one derived entirely from MIMIC and the other derived entirely from GICU. The optimised classifier is then retrained on the full training data (MIMIC and GICU), and its performance is tested on the held-out test data. This procedure is repeated over 100 random train-test splits to produce estimates of the mean and SD of classifier performances.

In order to determine the feature importances for each classifier, and therefore understand which features were most predictive of readiness-for-discharge, we calculated the permutation feature importance.[33] In short, this procedure involves iterative random permutation of the values of each feature, and the calculation of average loss of classifier performance (we used area under the receiver-operator-characteristic [ROC] curve) resulting from this feature randomisation. The overall performance of our trained classifiers, and the NLD criteria, was characterised by producing ROC and precision-recall (PRC) curves,[34] and by evaluating a suite of common performance metrics.

### RESULTS

The original specification of the NLD criteria proved to be highly conservative as expected, producing low false positive and true positive rates for both cohorts (online supplementary section D: tables 2–5). The true positive rates for MIMIC and GICU were 1.1% and 6.6%, respectively. Varying the threshold number of criteria required to make an RFD classification allowed us to produce ROC and PRC curves for the NLD criteria. These curves are illustrated in figure 1 for a single train-test data split. On this data split the NLD criteria obtained precisions of greater than 0.7 up to a recall of 0.6 for both cohorts. The

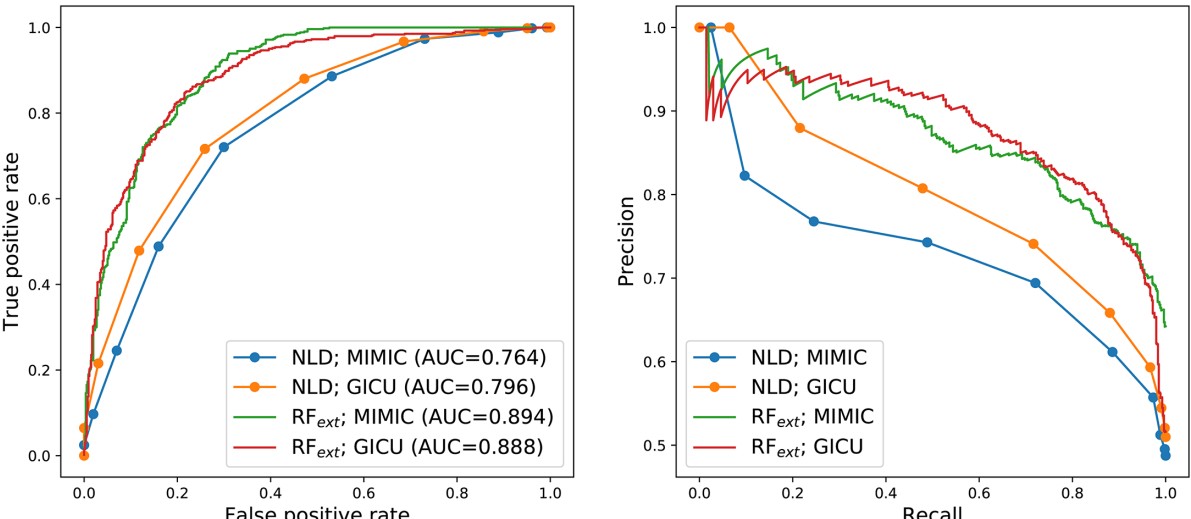

**Figure 1** Performance of the nurse-led discharge criteria and random forest with extended feature set (RF_ext) evaluated on held-out data for a single train-test split. Left: receiver-operator-characteristic curves with associated area-under-curve scores. Right: precision-recall curves. AUC, area-under-curve; GICU, general intensive care unit; NLD, nurse-led discharge; RF, random forest.

RF using the extended feature set showed large performance gains on this data split, with precisions of greater than 0.8 up to a recall of 0.8.

In general, the machine learning classifiers outperformed the NLD criteria. These performances, averaged over all 100 train-test data splits are summarised in table 3. The RF performed better than the LC on MIMIC,

according to all performance metrics, when using both the original and extended feature sets. On GICU the RF and LC produced similar scores. For this cohort, the LC with the original feature set narrowly outperformed the RF according to all metrics, but only won on three metrics (Area under receiver operating characteristic (AUROC); partialAUROC (pAUROC); Brier score)

| Table 3 | Performance metrics for the various classification systems | | | | | |
|---|---|---|---|---|---|---|
| | NLD | NLD_weighted | LC | RF | LC_extended | RF_extended |
| **GICU** | | | | | | |
| AUROC | 0.7913 (0.0098) | 0.8197 (0.0098) | 0.8788 (0.0087) | 0.8692 (0.0093) | **0.8822 (0.0091)** | 0.8721 (0.0094) |
| Accuracy | 0.7222 (0.0248) | 0.7829 (0.0339) | 0.8397 (0.0492) | 0.8389 (0.0496) | 0.8318 (0.0475) | **0.8426 (0.0505)** |
| F1 | 0.7473 (0.0109) | 0.7709 (0.0153) | 0.8109 (0.0099) | 0.8102 (0.0115) | 0.8050 (0.0119) | **0.8129 (0.0109)** |
| Specificity | 0.7000 (0.0000) | 0.7000 (0.0000) | 0.7000 (0.0000) | 0.7000 (0.0000) | 0.7000 (0.0000) | 0.7000 (0.0000) |
| pAUROC | 0.1469 (0.0061) | 0.1471 (0.0076) | 0.1961 (0.0068) | 0.1876 (0.0078) | **0.1989 (0.0068)** | 0.1888 (0.0079) |
| Brier | 0.2677 (0.0060) | 0.2265 (0.0083) | 0.1465 (0.0052) | 0.1502 (0.0056) | **0.1439 (0.0059)** | 0.1482 (0.0049) |
| Sensitivity | 0.7426 (0.0166) | 0.8098 (0.0263) | 0.8870 (0.0171) | 0.8860 (0.0196) | 0.8767 (0.0196) | **0.8909 (0.0185)** |
| **MIMIC** | | | | | | |
| AUROC | 0.7442 (0.0059) | 0.8248 (0.0056) | 0.8549 (0.0124) | 0.8605 (0.0122) | 0.8726 (0.0108) | **0.8859 (0.0110)** |
| Accuracy | 0.6783 (0.0125) | 0.8007 (0.0358) | 0.8366 (0.0513) | 0.8387 (0.0517) | 0.8494 (0.0533) | **0.8531 (0.0545)** |
| F1 | 0.6908 (0.0120) | 0.7830 (0.0103) | 0.8084 (0.0171) | 0.8097 (0.0158) | 0.8175 (0.0123) | **0.8201 (0.0133)** |
| Specificity | 0.7000 (0.0000) | 0.7000 (0.0000) | 0.7000 (0.0000) | 0.7000 (0.0000) | 0.7000 (0.0000) | 0.7000 (0.0000) |
| pAUROC | 0.1238 (0.0030) | 0.1429 (0.0043) | 0.1677 (0.0100) | 0.1729 (0.0099) | 0.1837 (0.0092) | **0.1955 (0.0091)** |
| Brier | 0.2510 (0.0029) | 0.1986 (0.0046) | 0.1470 (0.0065) | 0.1472 (0.0069) | 0.1394 (0.0056) | **0.1388 (0.0064)** |
| Sensitivity | 0.6713 (0.0126) | 0.8337 (0.0174) | 0.8827 (0.0282) | 0.8860 (0.0265) | 0.9001 (0.0207) | **0.9049 (0.0210)** |

All scores are averaged over 100 train-test data splits and given as: mean (SD). All metrics other than AUROC and Brier score are evaluated at a specificity of 0.7, using linear interpolation to estimate this operating point in receiver-operator-characteristic-space. NLD_weighted are the NLD criteria, weighted by feature importances from the LC. LC_extended and RF_extended are the machine learning classifiers with extended feature sets. Best scores for each metric are shown in bold.
GICU, general intensive care unit; LC, logistic classifier; NLD, nurse-led discharge; RF, random forest.

**Table 4** Feature importances given by the random forest (RF) and logistic classifier (LC), evaluated over 100 train-test data splits. Importance values are given as: mean (SD). Features are ranked according to mean importance value, and the table is ordered according to the ranking given by the LC

|  | Importance (LC) | Importance (RF) | Rank (LC) | Rank (RF) |
|---|---|---|---|---|
| gcs_min | 0.1053 (0.0026) | 0.1029 (0.0102) | 0 | 0 |
| airway | 0.0776 (0.0026) | 0.0602 (0.0076) | 1 | 1 |
| bun | 0.0190 (0.0009) | 0.0139 (0.0025) | 2 | 3 |
| fio2 | 0.0096 (0.0006) | 0.0205 (0.0024) | 3 | 2 |
| hr_max | 0.0063 (0.0009) | 0.0076 (0.0015) | 4 | 4 |
| haemoglobin | 0.0061 (0.0006) | 0.0040 (0.0014) | 5 | 6 |
| resp_max | 0.0037 (0.0006) | 0.0031 (0.0010) | 6 | 7 |
| hr_min | 0.0024 (0.0006) | 0.0047 (0.0014) | 7 | 5 |
| na | 0.0010 (0.0003) | 0.0005 (0.0004) | 8 | 15 |
| hco3 | 0.0009 (0.0003) | 0.0006 (0.0005) | 9 | 14 |
| spo2_min | 0.0005 (0.0002) | 0.0005 (0.0003) | 10 | 16 |
| bp_min | 0.0003 (0.0001) | 0.0013 (0.0009) | 11 | 11 |
| resp_min | 0.0001 (0.0001) | 0.0020 (0.0007) | 12 | 9 |
| pain | 0.0000 (0.0000) | 0.0009 (0.0006) | 13 | 13 |
| creatinine | 0.0000 (0.0000) | 0.0028 (0.0011) | 14 | 8 |
| k | 0.0000 (0.0000) | 0.0003 (0.0003) | 15 | 17 |
| temp_min | 0.0000 (0.0000) | 0.0012 (0.0009) | 16 | 12 |
| temp_max | 0.0000 (0.0000) | 0.0018 (0.0008) | 17 | 10 |

when the extended feature set was used. Overall, when training and testing on the imputed dataset, the RF with extended feature set showed the best performance. The complete case analysis (online supplementary file section D: table 6) produced qualitatively similar results but there was a clearer distinction between classifiers, with the LC performing better on GICU and the RF performing better on MIMIC. Average receiver operating characteristics are summarised for all classifiers in the online supplementary file section D: tables 7 and 8.

Broadly the two classifiers agreed as to which features were important in classifying patients as RFD (table 4). Eight of the features ranked in the top 10 by the LC were also ranked in the top 10 by the RF, and the Spearman's rank correlation coefficient between the feature rankings was 0.761 (p=0.00002). Both classifiers ranked *gcs_min* and *airway* as the two most important features by a significant margin. There was little change in these feature rankings under the complete case analysis (online supplementary file section D: table 9). We attempted to improve the classification performance of the NLD criteria by weighting each of the criteria according to the corresponding feature importance given by the logistic classifier. This weighting produced small performance gains over the original criteria (see NLD$_{weighted}$ in table 3), but not enough to warrant their use instead of a machine learning classifier in a clinical setting.

## DISCUSSION

Identifying which patients are suitable for ICU discharge is complex.[1 6] Delayed and out of hours discharges are associated with increased mortality[35] and patients in ICU who could be managed on the ward put an unnecessary strain on resources. The determination of RFD status is influenced by many unmeasured factors, such as ICU census,[25] and this leads to unwarranted variation in clinical decision making. Furthermore, the decision to declare someone fit for discharge is based on the judgement of individual clinicians and is likely to be given a lower priority than decisions about treatment options for patients who are more unwell.

In this study we have put forwards the concept of a decision support tool that would prompt clinicians to consider discharging a patient when they appear physiologically RFD. Such a prompt would occur by means of a dashboard notification, or 'nudge'.[20] It would need to be sufficiently sensitive as to recommend high numbers of potential discharges, while providing enough specificity to retain clinician engagement. Here we have detailed the development of two machine learning algorithms intended for such a purpose, and demonstrated their performance improvement over a previously published set of criteria that were originally aimed at discharge automation.[22] At a threshold specificity of 0.7, the algorithm with best overall performance achieved mean sensitivities of 0.8909 and 0.9049 for the GICU and MIMIC cohorts, respectively (online supplementary file section D: table

7). This represents a relatively high rate of false positives and suggests that further development is required before a tool based on this approach could be deployed clinically.

The features identified as most important by the classifiers were clinically meaningful. Clinicians will recognise that coma score; respiratory function and renal function are strongly related to successful ICU discharge. Under the LC certain features, such as body temperature and creatinine, appeared to be less important than we expected. This may be, in part, a consequence of patient heterogeneity on the GICU.[36] For example, body temperature may be predictive for patients with infection yet much of this predictive power is lost in our attempt to fit a general model for the whole ICU population. Similarly, although creatinine levels are indicative of renal function, persistently high creatinine should not be a criterion against discharge readiness for patients with chronic renal failure. The ability of the RF to better model such non-linear feature dependencies may explain why it gave a higher rank to these features.

In general, the performance of both classifiers would benefit from expanding the feature representation. The feature set we used was chosen to be directly analogous with the features tested by the discharge criteria. This feature set is restrictive, having been originally designed to be manually recorded by nurses using paper charts. We demonstrated that adding four extra features (age, sex, BMI and hours since admission) improved classification performance. However, machine learning methods have the power to further exploit the richness of the data held in electronic charting systems by including more physiological parameters, and learning the most predictive feature representation of these parameters.[37] One barrier to this approach is the challenge of harmonising the data, especially when combining data from different sources. This is one reason that we did not include diagnosis codes or severity of illness scores in this study, although they have previously been shown to be predictive of adverse events following discharge.[11 12] During a patient's stay in ICU, many of their physiological parameters are controlled by clinical intervention, and their expected physiological state is dependent on their medical history (see, eg, guidelines on acceptable levels of haemoglobin in different patient types).[38] Therefore, conditioning features on medical interventions and applying methods for patient sub-typing[36 39] are two improvements that we expect could significantly boost performance. Also, although the complete case analysis did not qualitatively alter our results, the development of a more sophisticated multiple-imputation strategy[40] would likely improve performance by making best use of the available training data and exploiting the value in missingness.[41]

A range of different tools and methods have previously been proposed with the aim of improving ICU discharge practice. These tools range from criteria to evaluate discharge readiness,[22 42] to guidelines for discharge planning and education.[6] Additionally, a number of risk models have been developed to predict adverse outcomes following ICU discharge.[11–13 15 43] In particular Badawi and Breslow demonstrated that mortality and readmission should be modelled independently as separate outcomes.[12] Clearly a comparative evaluation of the existing tools is required in a clinical setting. We argue that a future decision support system for discharge planning should draw elements from all these methods. A screening algorithm, such as the one we have outlined here, could notify clinicians of dischargeable patients in cases where the decision is clear. Decisions around high-risk patients, which are frequently required, would benefit from an extra level of decision support, such as individual predictions of mortality and readmission risk.[12] The increasing availability of intensive care research datasets[44 45] is sure to improve the performance and generality of such models, particularly as methods from transfer learning are applied.[15] Ultimately the benefit from these models comes from the manner in which they are deployed. We have shown in previous work that subtle changes to the presentation of information can have significant impact on clinical decision-making.[20] The aggregate effects of the small improvements produced by such approaches could be widely beneficial.[46] We suggest that the proposed decision support system would maximise engagement by addressing issues of model interpretability,[47 48] and could leverage clinical expertise by learning online with a human-in-the-loop.[49]

## CONCLUSION

This work outlines a framework for the use of machine learning algorithms to identify patients that are physiologically fit for discharge from the ICU. A decision support tool based on these methods could contribute to the solution of this significant clinical and operational problem by streamlining the discharge process and reducing unnecessary ICU stay. We have identified a number of improvements that would be required before the deployment and testing of such a tool in a clinical setting, and highlighted how the tool would benefit from the inclusion of multiple complementary modelling frameworks. As more patient data becomes available in the wider hospital setting there is extensive scope to use data-driven methods, such as the one presented here, to improve patient flow through hospitals.

**Acknowledgements** We would like to thank Graeme Palmer, Amy Weaver and Russell McDonald-Bell for their support in accessing and understanding the GICU data.

**Contributors** CJM is the main author and conducted the data processing and analysis. RS-R and DJL made important technical and methodological contributions. IDG, AC and CPB drove the study concept and design. The clinical expertise of THG, MJCT and CPB informed all stages of the project, in particular study design and interpretation of results. All authors contributed to writing the manuscript and approved the final version.

**Funding** This work was supported in part by EurValve (Personalised Decision Supportfor Heart Valve Disease), Project Number: H2020 PHC-30-2015, 689617. CJM was funded by the Elizabeth Blackwell Institute Catalyst Fund. DJL is funded by Wellcome Trust and Royal Society Grant Number WT104125MA.

**Competing interests** None declared.

**Patient consent for publication** Not required.

**Ethics approval** CAG guidelines followed and study protocol presented to University Hospitals Bristol NHS Foundation Trust institutional review board.

**Provenance and peer review** Not commissioned; externally peer reviewed.

**Data sharing statement** Feature matrices will be made available on Dryad. Python code for analysis and processing on request directly from the corresponding author.

**Open access** This is an open access article distributed in accordance with the Creative Commons Attribution 4.0 Unported (CC BY 4.0) license, which permits others to copy, redistribute, remix, transform and build upon this work for any purpose, provided the original work is properly cited, a link to the licence is given, and indication of whether changes were made. See: https://creativecommons.org/licenses/by/4.0/.

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
