## [Reviewer comments · BMJ Open]

ARTICLE DETAILS

TITLE (PROVISIONAL)	Towards a decision support tool for intensive care discharge: machine learning algorithm development using electronic health care data from MIMIC-III and Bristol, UK.
AUTHORS	McWilliams, Christopher; Lawson, Daniel; Santos-Rodriguez, Raul; Gilchrist, Iain; Champneys, Alan; Gould, Timothy; Thomas, Mathew; Bourdeaux, Christopher

VERSION 1 – REVIEW

REVIEWER	Reviewer name: Leo Anthony Celi Institution and Country: Massachusetts Institute of Technology, United States Competing interests: None declared
REVIEW RETURNED	29-Aug-2018

GENERAL COMMENTS	This paper lays the groundwork for promising advances that will move complex decisions from the domain of subjective clinical intuition toward a more objective data-substantiated insight. At present, the timing and the disposition of an ICU discharge are based solely on heuristics. But while a decision support tool using available data to support clinicians for this purpose is an interesting idea, ICU clinicians may not feel that they need such a tool. The authors suggest that the tool could be employed to standardize and optimize (in terms of timing) the ICU discharge process and yield some measurable improvements. We suggest explicitly describing a use case to illustrate where the improvements will come from. As an example, a patient deemed at high-risk of deterioration is transferred to a telemetry bed, while another is deemed stable for the medical ward. Misclassification by the clinician or the team would result in wasted resources in the former, and a risky premature de-escalation of care in the latter. Although we are convinced of the value of the approach that is presented in the paper, we have significant concerns about this work as it is currently presented. Foremost, we believe that the authors did not present a comprehensive literature search around this topic. There are several papers that are quite similar to the concept of ICU discharge readiness that should have been cited: 1. Badawi O, Breslow MJ. Readmissions and death after ICU discharge: development and validation of two predictive models. PLoS One. 2012;7(11):e48758.2. Badawi O, Liu X, Hassan E, Amelung PJ, Swami S. Evaluation of ICU Risk Models Adapted for Use as Continuous Markers of Severity of Illness Throughout the ICU Stay. Crit Care Med. 2018 Mar;46(3):361-367.
---

3. Rojas JC, Carey KA, Edelson DP, Venable LR, Howell MD, Churpek MM. Predicting Intensive Care Unit Readmission with Machine Learning Using Electronic Health Record Data. *Ann Am Thorac Soc.* 2018 Jul;15(7):846-853.
4. Cosgriff CV, Celi LA, Sauer CM. Boosting Clinical Decision-making: Machine Learning for Intensive Care Unit Discharge. *Ann Am Thorac Soc.* 2018 Jul;15(7):804-805.

In terms of the methods employed, we raise the following issue:

1. A relatively small number of features were extracted. And while feature selection may be completely heuristics-driven as performed in this paper, the power of machine learning lies in its ability to take advantage of large sets of features. An approach that has gained traction is a combination computer- and domain expert-driven feature selection that we recommend.

2. We recommend the inclusion of admission metrics such as diagnosis and illness severity, and demographic information in the model. Prior studies suggest they are important features in assessing risk of ICU readmission at the time of discharge.

3. The negative outcomes were readmission or death at any time after ICU discharge during the hospital admission. It may be 'unfair' to attribute death or readmission more than 48 hours after ICU discharge to a faulty discharge process. It might be advisable to separate death from readmission as according to the results of Badawi and Breslow (ref above), death is a much more dependably predictable phenomenon in this context.

4. The primary analysis excludes subjects with missing data under the assumption of uninformative "missingness". Although this is appropriately addressed with a sensitivity analysis, it may make more sense to include the imputed dataset in the main analysis.

5. The use of cross validation is appropriate, but the description in the methods is confusing. The text notes 100 data splits were performed. Was 100-fold cross validation pursued? The percentage of the total data saved as a held out test set and the amount of folds used in cross validation would make this portion of the paper more clear.

6. The reporting of ROC and PRC curves is also appropriate, although the choice of partial AUC is questionable and has been questioned (*Stat Med.* 2005 Jul 15;24(13):2025-40.) Furthermore the performance results are not presented clearly.

7. Were the data were normalized? This does not appear to be explicitly mentioned. The use of LASSO logistic regression necessitates centering and scaling the data, and the use of coefficient as measures of feature importance would only be interpretable under these circumstances.

8. Finally, a comparison of feature importance between the Gini approach and the coefficients of a regression is of questionable validity

In summary, this work is addressing a real and relevant clinical problem, but fails to take into account prior work in this area, employs methods that raise questions, and insufficiently explains the approaches and why such approaches were chosen.

	LINE by LINE: P2, L50 – this approach does not constitute “the first step towards a decision support tool that would help clinicians identify dischargeable patients as early as possible.” What has been done here is a technically different approach to the same clinical issue. Introduction P4, L4 - This should read, “when this resource is restrained”. P4, L6 - ‘as soon’ is wrongly repeated P4, L7 - not just emergency patients- also the performance of elective and urgent surgeries that require ICU care P4, L9 - There are other risks to remaining in the ICU unnecessarily when patients are deemed ready for discharge. The risk of iatrogenic harm results from a culture of over-testing and over-treatment. P5, L7 – We are unfamiliar with the practice of nurses performing ICU discharges. We suggest elaborating on this in the text. P5, L28 - The need for a hemoglobin of ≥ 9 is not evidence-based and requires some explanation. P5, L32 - The persistent high creatinine for patients with chronic renal failure/ESRD should not be a criterion against discharge readiness. P5, L49 - How was the subset of MIMIC patients selected? This should appear in the main manuscript. P11, L52 – ‘Vasopressors’ is misspelled. P13, L5 – We recommend changing ‘solved’ to ‘attempt to solve’.
--	---

REVIEWER	Reviewer name: Ari Ercole Institution and Country: University of Cambridge, UK Competing interests: None declared
REVIEW RETURNED	22-Sep-2018

GENERAL COMMENTS	The authors attempt to construct a model to predict whether an ICU patient is fit for discharge. They demonstrate that their models outperform a previously cited nursing tool. Their approach is not without merit and is broadly speaking well conducted although I'm unclear that it has sufficient performance or refinement to be used in its present form (the authors appropriately make no such claim but may wish to reflect/comment on what would need to be done to make this a more useful tool). In the introduction the authors strongly imply that decision support in this arena "...would reduce the cognitive load of clinicians and free up scarce resource...". This is a bold statement and is difficult to justify if the system is imperfect (which it will always be). For example, a system with a high sensitivity but low specificity would generate many 'false alarms' which would either lead to dangerous discharge or, more likely, need another tier of assessment which would in fact increase workload. This statement should be toned-down. The authors have chosen to predict whether a patient will fulfil the criteria given by Knight for the featureset design. This is reasonable but the authors should comment on whether a different (more or less parsimonious) model might be better or potentially feasible.. The technique used to balancing the dataset has merit but needs further justification. It is not clear that the conditions 8 days prior to the decision point will be clinically and predictively homogeneous.
--

	Indeed there may be important trend information that is not captured in the simple time-insensitive models used. It would at least be reassuring to see a histogram of the day number of (discharge - 8) days. It may not be appropriate to include patients very close to the admission day in the NRFD group- we may expect these to be rather different in characteristics. At very least, the sensitivity to this assumption should be examined. An alternative approach (although I suspect inferior) would be to simply re-weight the NRFD in the model. Another alternative, which may be better, would be to additionally include days post admission in the model to attempt to account for temporal trends. The use of 24hour periods prior to callout is probably an appropriate strategy but needs some more justification. For example, this subsequently means that daily bloods (likely done in the morning) are now no longer at the same point in each 24hour period. There may be many other factors that dictate a decision to discharge from ICU other than medical status or progression (likely ward staffing and ability to fulfil patient care needs). The authors should therefore clarify what role such a tool should have as this will influence the likely sensitivity and specificity that is desirable (i.e. if this is a screening tool then a high sensitivity is required). The authors have appropriately checked their model against an imputed dataset. However there is likely to be error from the imputation- have they considered a multiple imputation strategy and evaluating the imputation variation? I understand the motivation for using areas under the partial AUC curves but isn't the choice of FPR = 0.3 rather arbitrary? Furthermore, since the statistics of pAUCs are not so frequently reported, it makes the authors work difficult to compare to general sorts of performance that might be obtained in the medical prediction literature. I think that I would like to see the overall AUCs and perhaps some evaluation of what sort of FPRs would be obtained under various choices of threshold (i.e. what sort of trade-offs are possible). The authors should probably cite some additional metrics of performance in the main text. The authors are not entirely correct in the claim regarding demonstrating the feasibility of transfer learning in their paper: We have previously published a machine learning model for predicting unexpected readmission which used a transfer learning methodology (see Desautels et al. BMJ Open 2017;7:e017199. doi:10.1136/bmjopen-2017-017199). There may be others.
--	---

VERSION 1 – AUTHOR RESPONSE

Reviewer: 1

Reviewer Name: Leo Anthony Celi

Institution and Country: Massachusetts Institute of Technology, United States

Please state any competing interests or state 'None declared': None declared

Please leave your comments for the authors below

This paper lays the groundwork for promising advances that will move complex decisions from the domain of subjective clinical intuition toward a more objective data-substantiated insight. At present, the timing and the disposition of an ICU discharge are based solely on heuristics.

But while a decision support tool using available data to support clinicians for this purpose is an interesting idea, ICU clinicians may not feel that they need such a tool. The authors suggest that the tool could be employed to standardize and optimize (in terms of timing) the ICU discharge process and yield some measurable improvements. We suggest explicitly describing a use case to illustrate where the improvements will come from. As an example, a patient deemed at high-risk of deterioration is transferred to a telemetry bed, while another is deemed stable for the medical ward. Misclassification by the clinician or the team would result in wasted resources in the former, and a risky premature de-escalation of care in the latter.

We have now outlined, in the introduction, the envisaged use case for a decision support tool that would use a ready for discharge classifier. This is expanded on in the revised discussion (paragraphs 2 and 5).

From the introduction:

In our previous work on the psychology of clinical decision making we have demonstrated the effectiveness of simple ‘nudge’ based interventions in changing clinical practice[19–21]. Building on this foundation we were motivated to develop a classifier to automatically flag patients that appear physiologically fit for discharge. The intention is that such a screening tool could streamline morning ward rounds by allowing staff to focus their attention on the most likely-dischargeable patients. The tool could also prompt clinicians to consider discharge decisions at other times of day, outside of normal rounds.

Although we are convinced of the value of the approach that is presented in the paper, we have significant concerns about this work as it is currently presented.

Foremost, we believe that the authors did not present a comprehensive literature search around this topic. There are several papers that are quite similar to the concept of ICU discharge readiness that should have been cited:

1. Badawi O, Breslow MJ. Readmissions and death after ICU discharge: development and validation of two predictive models. *PLoS One*. 2012;7(11):e48758.
2. Badawi O, Liu X, Hassan E, Amelung PJ, Swami S. Evaluation of ICU Risk Models Adapted for Use as Continuous Markers of Severity of Illness Throughout the ICU Stay. *Crit Care Med*. 2018 Mar;46(3):361-367.
3. Rojas JC, Carey KA, Edelson DP, Venable LR, Howell MD, Churpek MM. Predicting Intensive Care Unit Readmission with Machine Learning Using Electronic Health Record Data. *Ann Am Thorac Soc*. 2018 Jul;15(7):846-853.
4. Cosgriff CV, Celi LA, Sauer CM. Boosting Clinical Decision-making: Machine Learning for Intensive Care Unit Discharge. *Ann Am Thorac Soc*. 2018 Jul;15(7):804-805.

Thank you for bringing these to our attention. They have been read, and cited appropriately in the introduction, and referred to in the discussion where relevant.

In terms of the methods employed, we raise the following issue:

1. A relatively small number of features were extracted. And while feature selection may be completely heuristics-driven as performed in this paper, the power of machine learning lies in its ability to take advantage of large sets of features. An approach that has gained traction is a combination computer- and domain expert-driven feature selection that we recommend.

We acknowledge that the small number of features restricts the benefits of our machine learning approach. The choice of features was originally guided by our desire to produce a direct comparison of the baseline performance provided by Knight's discharge criteria. In future work we certainly intended to exploit the richness of the available EHR data more fully.

From the discussion:

However, machine learning methods have the power to further exploit the richness of the data held in electronic charting systems by including more physiological parameters, and learning the most predictive feature representation of these parameters (see for example [34]).

2. We recommend the inclusion of admission metrics such as diagnosis and illness severity, and demographic information in the model. Prior studies suggest they are important features in assessing risk of ICU readmission at the time of discharge.

We agree, and had intended to do this in future work (see above response). We have now added four some demographic variables to our classifiers (age, sex, bmi, hours since admission). Unfortunately, we have not been able to include diagnosis or severity of illness features at this stage. This is due to differences in the datasets. Our diagnosis codes follow the ICNARC coding model, while those in MIMIC are ICD9. Our precomputed severity of illness scores are ICNARC specific, and are used to generate standardised morality ratios for audit. To include diagnosis and severity of illness features would have required more data processing than we had time for before the resubmission deadline. However, we acknowledge that it would likely improve our modelling and is desirable for future work. This is commented on in the discussion section:

This is one reason that we did not include diagnosis codes or severity of illness scores in this study, although they have previously been shown to be predictive of adverse events following discharge[11,12].

3. The negative outcomes were readmission or death at any time after ICU discharge during the hospital admission. It may be 'unfair' to attribute death or readmission more than 48 hours after ICU discharge to a faulty discharge process. It might be advisable to separate death from readmission as according to the results of Badawi and Breslow (ref above), death is a much more dependably predictable phenomenon in this context.

Both of these points are now commented on in the manuscript (reproduced below). Our modelling is somewhat different from e.g. Badawi and Breslow in that these negative outcomes only form a small part of the negative (NRF) class that we are training on. The bulk of this class is made up of patients sampled at earlier time points in their stay, when they unambiguously require further critical care. This difference in the modelling framework was not explicitly stated in the previous version of the manuscript, and we hope it is now clearer. We also comment, in our revised discussion, on the need for a good decision support tool for discharge planning to incorporate multiple risk models of different outcomes.

From the Methods:

We note that it is more conventional to use readmission (or mortality) within 48 hours to define a negative outcome related to ICU care[12,26]. However, this practice is not universal[27] and in our case it was not possible because of limitations in the data available locally.

From the Discussion:

Additionally, a number of risk models have been developed to predict adverse outcomes following ICU discharge[11–13,15,43]. In particular Badawi and Breslow demonstrated that mortality and readmission should be modelled independently as separate outcomes[12].

4. The primary analysis excludes subjects with missing data under the assumption of uninformative “missingness”. Although this is appropriately addressed with a sensitivity analysis, it may make more sense to include the imputed dataset in the main analysis.

This is now switched as recommended – the results in the main text use the imputed dataset, and the complete case analysis has been moved to the SI.

5. The use of cross validation is appropriate, but the description in the methods is confusing. The text notes 100 data splits were performed. Was 100-fold cross validation pursued? The percentage of the total data saved as a held out test set and the amount of folds used in cross validation would make this portion of the paper more clear.

We believe it was a combination of unclear description and a non-conventional training methodology which led to confusion here. We have redone the analysis using (2-fold) multiple-source cross-validation*. This is a slight change from our initial methodology, which was effectively single-fold multiple-source cross-validation. Essentially the training and cross-validation folds consist of data from only a single source (MIMIC OR GICU), so the source do not get mixed up. Hopefully the description is now clear in manuscript.

*As in <http://proceedings.mlr.press/v28/geras13.pdf>

6. The reporting of ROC and PRC curves is also appropriate, although the choice of partial AUC is questionable and has been questioned (Stat Med. 2005 Jul 15;24(13):2025-40.) Furthermore the performance results are not presented clearly.

We now provide a number of different metrics to give a more general picture of performance that is comparable with other studies (table 3). pAUC is still in there but is no longer the sole metric. We retain the ROC and PRC curves. We also have included tables in the supplementary online file (SI: tables 8 and 9) that were requested by reviewer #2, characterising trade-offs in performance.

7. Were the data were normalized? This does not appear to be explicitly mentioned. The use of LASSO logistic regression necessitates centering and scaling the data, and the use of coefficient as measures of feature importance would only be interpretable under these circumstances.

Data were standardised. This is now clearly stated at the end of subsection “Feature extraction” in the Methods section:

When training and testing the machine learning classifiers, features were standardised by subtracting the mean and dividing by the standard deviation.

8. Finally, a comparison of feature importance between the Gini approach and the coefficients of a regression is of questionable validity

We acknowledge that the feature importance values produced by these two different approaches are not directly comparable (although we would argue that there is still meaning in comparing the rankings). We have changed our approach to use the permutation feature importance i.e. mean decrease in AUROC when a feature is randomly permuted. This gives a feature importance measure that is directly comparable between classifiers, and produces similar results to the original method (table 4).

From the Methods:

In order to determine the feature importances for each classifier, and therefore understand which features were most predictive of readiness-for-discharge, we used the permutation feature importance[33]. In short this procedure involves iterative random permutation of the values of each feature, and the calculation of average loss of classifier performance (we used area under the ROC curve) resulting from this feature randomisation.

LINE by LINE:

P2, L50 – this approach does not constitute “the first step towards a decision support tool that would help clinicians identify dischargeable patients as early as possible.” What has been done here is a technically different approach to the same clinical issue.

Introduction

We have replaced this statement.

P4, L4 - This should read, “when this resource is restrained”.

We have replaced with “when the resource is constrained”. We feel that the term “constrained” captures the meaning that we are trying to convey here, which is that units across the NHS are operating with severely restricted resources.

P4, L6 - ‘as soon’ is wrongly repeated

Thanks! Repetition removed.

P4, L7 - not just emergency patients- also the performance of elective and urgent surgeries that require ICU care

Agreed. ‘Emergency’ removed.

P4, L9 - There are other risks to remaining in the ICU unnecessarily when patients are deemed ready for discharge. The risk of iatrogenic harm results from a culture of over-testing and over-treatment.

Amended to reflect this.

P5, L7 – We are unfamiliar with the practice of nurses performing ICU discharges. We suggest elaborating on this in the text.

This is now clarified in the text – by additions to the introduction and the methods section.

e.g. from the Methods:

The nurse-led discharge (NLD) criteria proposed by Knight[22] consist of a set of constraints on various routinely collected vital signs and laboratory results. If a patient meets all the criteria for a period of at least four hours, Knight states that they may be safely discharged by a nurse. The motivation behind developing these criteria was to facilitate discharge by nurses in cases where the decision was clear, and there is some evidence of improved bed allocation when using such a nurse-led system[22,24,25].

P5, L28 - The need for a hemoglobin of ≥ 9 is not evidence-based and requires some explanation.

This was the value used in the original criteria published by Knight. We acknowledge that this is not evidence based.

We comment in the discussion:

During a patient's stay in ICU, many of their physiological parameters are controlled by clinical intervention, and their expected physiological state is dependent on their medical history (see, for example, guidelines on acceptable levels of Hb in different patient types[37]).

And in section A of the supplementary file:

We note here that the criterion value of 9g/dL, above which haemoglobin is deemed acceptable, does not appear to be evidence based. We refer the reader to this discussion [2] on haemoglobin in critical care.

P5, L32 - The persistent high creatinine for patients with chronic renal failure/ESRD should not be a criterion against discharge readiness.

Absolutely. More generally the simplistic features do not capture trends and are not conditioned on medication or medical history. Hence the call for a more sophisticated approach. See our revised discussion.

P5, L49 - How was the subset of MIMIC patients selected? This should appear in the main manuscript.

Have updated subsection 'Cohort selection' to include this:

The second data source was derived from the MIMIC-III database[23], from which we selected patients who were admitted to medical or surgical intensive care since this approximates the patient type in GICU. We restricted our analysis to the 'Metavision' subset of MIMIC-III, since the labelling of the variables required to evaluate the NLD criteria was found to be more consistent in this portion of the database. Furthermore, we selected only the first intensive care stay of any given hospital admission, and only those stays for which there was a recorded callout (ready-for-discharge) time. Following these criteria, we arrived at a subset of 7592 patients from MIMIC-III, forming the cohort we refer to hereafter as MIMIC.

P11, L52 – 'Vasopressors' is misspelled.

Thanks! Although vasopressor comment now removed.

P13, L5 – We recommend changing 'solved' to 'attempt to solve'.

Conclusion revised accordingly.

Reviewer: 2

Reviewer Name: Ari Ercole

Institution and Country: University of Cambridge, UK

Please state any competing interests or state 'None declared': None declared

Please leave your comments for the authors below

The authors attempt to construct a model to predict whether an ICU patient is fit for discharge. They demonstrate that their models outperform a previously cited nursing tool.

Their approach is not without merit and is broadly speaking well conducted although I'm unclear that it has sufficient performance or refinement to be used in its present form (the authors appropriately make no such claim but may wish to reflect/comment on what would need to be done to make this a more useful tool).

We agree that there is more to be done to make this a useful tool. We feel that this is addressed in the revised discussion.

In the introduction the authors strongly imply that decision support in this arena "...would reduce the cognitive load of clinicians and free up scarce resource...". This is a bold statement and is difficult to justify if the system is imperfect (which it will always be). For example, a system with a high sensitivity but low specificity would generate many 'false alarms' which would either lead to dangerous discharge or, more likely, need another tier of assessment which would in fact increase workload. This statement should be toned-down.

This statement refers to the possibility of simple task automation and decision support systems in general. We think that such technologies do have the potential to reduce cognitive load. We acknowledge that the method we present is a long way off achieving this (see above).

From introduction (slightly amended):

The availability of such routinely collected patient data presents the opportunity to apply methods from data science, with the potential to transform healthcare in a number of ways[7,8]. Two particular avenues for development are the automation of simple tasks[9] and the implementation of decision support systems[10], both of which could reduce the cognitive load of clinicians and free up scarce resource for tasks that require human expertise.

The authors have chosen to predict whether a patient will fulfil the criteria given by Knight for the featureset design. This is reasonable but the authors should comment on whether a different (more or less parsimonious) model might be better or potentially feasible..

Agreed. In revised discussion we comment on the need for development of less parsimonious models (i.e. more features), but also comment on the value of including multiple complementary modelling frameworks into a decision support system (citing e.g. SWIFT as one simpler model for this appropriate for this task).

From the discussion:

However, machine learning methods have the power to further exploit the richness of the data held in electronic charting systems by including more physiological parameters, and learning the most predictive feature representation of these parameters (see for example [36])

The technique used to balancing the dataset has merit but needs further justification. It is not clear that the conditions 8 days prior to the decision point will be clinically and predictively homogeneous. Indeed there may be important trend information that is not captured in the simple time-insensitive models used. It would at least be reassuring to see a histogram of the day number of (discharge - 8) days. It may not be appropriate to include patients very close to the admission day in the NTFD group- we may expect these to be rather different in characteristics. At very least, the sensitivity to this assumption should be examined. An alternative approach (although I suspect inferior) would be to simply re-weight the NTFD in the model. Another alternative, which may be better, would be to additionally include days post admission in the model to attempt to account for temporal trends.

In the new analysis we have removed patients from very early in their stay (within first 24 hours) from the re-sampling. We have also, on the advice of reviewer #1, included current length of stay (in hours) as a feature. We also acknowledge in the discussion the methods to capture temporal trends in variables (e.g. improvement in creatinine since admission versus snapshot of creatinine value) would be highly desirable and likely improve performance.

The use of 24hour periods prior to callout is probably an appropriate strategy but needs some more justification. For example, this subsequently means that daily bloods (likely done in the morning) are now no longer at the same point in each 24hour period.

We have expanded the justification in the SI, which is reference from the main text

From the Methods:

To do this we sampled patients at between three and eight days prior to their callout (see supplementary section B: figures 1-3), under the assumption that patients were not-ready-for-discharge at this point in time, regardless of their eventual outcome state (positive or negative). Patients within the first 24 hours of their ICU stay were omitted from this sample. Full details of the sampling procedure are given in section B of the online supplementary file.

From section B.2 of the supplementary file:

The reason for sampling in exact multiples of 24 hours prior to callout was to ensure the removal of time of day effects from the data (see section B.3).

An alternative strategy would be to sample and classify patients at a set time point each day. This strategy would have the benefit of preserving the time relationship of certain daily activities. For example, daily bloods are likely done in the morning. Our strategy, of sampling in 24 hour multiples prior to callout may introduce more variability in the time since last bloods, and possibly other measurements/interventions. However, given that we wanted to work towards a classifier that is not constrained by activities (such as discharge decisions) occurring at fixed times of day, we feel that the chosen sampling strategy is justified.

There may be many other factors that dictate a decision to discharge from ICU other than medical status or progression (likely ward staffing and ability to fulfil patient care needs). The authors should therefore clarify what role such a tool should have as this will influence the likely sensitivity and specificity that is desirable (i.e. if this is a screening tool then a high sensitivity is required).

Our envisaged use case for such a tool has now been clarified in the introduction, and further in the discussion.

The authors have appropriately checked their model against an imputed dataset. However there is likely to be error from the imputation- have they considered a multiple imputation strategy and evaluating the imputation variation?

This is commented in the revised discussion as an improvement for future work.

From the discussion:

Also, although the complete case analysis did not qualitatively alter our results, the development of a more sophisticated multiple-imputation strategy[40] would likely improve performance by making best use of the available training data and exploiting the value in missingness[41].

I understand the motivation for using areas under the partial AUC curves but isn't the choice of FPR = 0.3 rather arbitrary? Furthermore, since the statistics of pAUCs are not so frequently reported, it makes the authors work difficult to compare to general sorts of performance that might be obtained in the medical prediction literature. I think that I would like to see the overall AUCs and perhaps some evaluation of what sort of FPRs would be obtained under various choices of threshold (i.e. what sort of trade-offs are possible). The authors should probably cite some additional metrics of performance in the main text.

We have added a suite of performance metrics in order to give a more complete characterisation or performance, that is comparable with other studies (table 3). We have also produced tables to illustrate the trade-offs as requested (it gives the sensitivities at different FPRs). These tables are attached here and reproduced in the supplementary file (SI: tables 8 and 9).

The authors are not entirely correct in the claim regarding demonstrating the feasibility of transfer learning in their paper: We have previously published a machine learning model for predicting unexpected readmission which used a transfer learning methodology (see Desautels et al. BMJ Open 2017;7:e017199. doi:10.1136/bmjopen-2017-017199). There may be others.

Thank you for drawing our attention to this work. We have amended our discussion accordingly. (We also note that, although we are learning from multiple source domains to produce general classifiers, we have not explicitly employed transfer learning methods. Your study is the first we know of in this area).

From the discussion:

The increasing availability of intensive care research datasets[44,45] is sure to improve the performance and generality of such models, particularly as methods from transfer learning are applied[15].

VERSION 2 – REVIEW

REVIEWER	Reviewer name: Leo Anthony Celi Institution and Country: Massachusetts Institute of Technology Competing interests: None
REVIEW RETURNED	22-Nov-2018

GENERAL COMMENTS	The authors have adequately addressed our comments and concerns.
--

REVIEWER	Reviewer name: Ari Ercole Institution and Country: University of Cambridge Division of Anaesthesia
-----------------	---

	Competing interests: None declared
REVIEW RETURNED	07-Dec-2018

GENERAL COMMENTS	The authors have satisfactorily addressed my concerns either by improving the analysis / presentation or by acknowledging limitations more fully. There is now sufficient information for a reader to make an informed judgement about the work and I would therefore be happy for this work to be published from a technical point of view.
--